# Impact of Endurance Training on Regeneration of Axons, Glial Cells, and Inhibitory Neurons after Spinal Cord Injury: A Link between Functional Outcome and Regeneration Potential within the Lesion Site and in Adjacent Spinal Cord Tissue

**DOI:** 10.3390/ijms24108616

**Published:** 2023-05-11

**Authors:** Katarina Kiss Bimbova, Maria Bacova, Alexandra Kisucka, Ján Gálik, Maria Ileninova, Tomas Kuruc, Martina Magurova, Nadezda Lukacova

**Affiliations:** Department of Neurodegeneration, Plasticity and Repair, Institute of Neurobiology, Biomedical Research Centre of Slovak Academy of Sciences, Soltesovej 4-6, 040 01 Kosice, Slovakia; bacova@saske.sk (M.B.); kisucka@saske.sk (A.K.); galik@saske.sk (J.G.); ileninova@saske.sk (M.I.); kuruc@saske.sk (T.K.); magurova@saske.sk (M.M.)

**Keywords:** spinal cord compression, endurance training, regeneration of cell populations, neurological outcome

## Abstract

Endurance training prior to spinal cord injury (SCI) has a beneficial effect on the activation of signaling pathways responsible for survival, neuroplasticity, and neuroregeneration. It is, however, unclear which training-induced cell populations are essential for the functional outcome after SCI. Adult Wistar rats were divided into four groups: control, six weeks of endurance training, Th9 compression (40 g/15 min), and pretraining + Th9 compression. The animals survived six weeks. Training alone increased the gene expression and protein level of immature CNP-ase oligodendrocytes (~16%) at Th10, and caused rearrangements in neurotrophic regulation of inhibitory GABA/glycinergic neurons at the Th10 and L2 levels, known to contain the interneurons with rhythmogenic potential. Training + SCI upregulated markers for immature and mature (CNP-ase, PLP1) oligodendrocytes by ~13% at the lesion site and caudally, and increased the number of GABA/glycinergic neurons in specific spinal cord regions. In the pretrained SCI group, the functional outcome of hindlimbs positively correlated with the protein levels of CNP-ase, PLP1, and neurofilaments (NF-l), but not with the outgrowing axons (Gap-43) at the lesion site and caudally. These results indicate that endurance training applied before SCI potentiates the repair in damaged spinal cord, and creates a suitable environment for neurological outcome.

## 1. Introduction

Regular physical activity is a powerful physiological stimulus known to prevent and even treat various age-related chronic diseases, mental health [1], neurodegenerative diseases [2], and to reduce the burden of the diseasome of physical inactivity [3]. Since the benefits of circulating bioactive molecules were shown to be specifically regulated by both acute and regular exercise [4,5], the global target of the World Health Organization (WHO) is to reduce physical inactivity by 10% by 2025, and by 15% by 2030 (WHO global action plan on physical activity 2018–2030) [6]. Currently, regular exercise is the most effective noninvasive post-SCI therapeutic approach, due, in part, to the fact that it increases the release of neurotrophic factors [7]. It is no wonder that growth factors, with their ability to promote cell survival and axonal growth, became attractive agents used to address regeneration following SCI [8,9].

Growth factors (GFs) are of great interest for their effect on intracellular signaling and regenerative potential. Physiological signals regulated by GFs are essential for controlling several cellular processes, including proliferation and differentiation. Like most other biologically active polypeptides, the synthesis of neurotrophins occurs in the rough endoplasmic reticulum, where the precursors—proneurotrophins—are packaged into secretory vesicles. They are secreted mainly by neurons, astrocytes, and microglial cells; however, they can also be found in tissues outside the nervous system (kidney, blood platelets, retina, etc.) [10,11]. GFs bind on specific cell-surface receptors that subsequently transmit their signals to intracellular components, and this event results in altered gene expression [12]. Neurotrophins binding to tropomyosin receptor kinase (Trk) receptors ultimately lead to the activation of specific signaling cascades. Several signaling pathways known to be activated are phosphatidylinositol 3-kinase/protein kinase B (PI3K/Akt), Ras/mitogen-activated protein kinase (Ras/MAPK), and phospholipase C-γ (PLC-γ). These cascades play well-known roles in synaptic plasticity, neuronal survival, and axonal outgrowth [13,14]. In recent years, the number of studies reporting on the positive effect of GFs applied exogenously and/or endogenously (noninvasive neurotrophic support by physical activity/exercise) is growing exponentially [15,16,17]. These effects, together with additional actions in altering the physiology of neurons and their synapses, show a strong potential for their roles in spinal cord repair processes. Since most SCIs have a negative impact on functional outcome, any therapy that enhances synaptic and cellular function is potentially useful [8,13].

It is generally accepted that most descending and peripheral afferent systems project into spinal interneurons rather than to motoneurons [18]. SCI interrupts important neuronal pathways between the supraspinal centers and spinal cord, and very often lead to the loss of neurons providing presynaptic inhibition to alpha motoneurons [19,20]. The identification of inhibitory neurons participating in the transmission of these impulses, and the mechanisms of their regeneration after SCI, is therefore extremely important. Our recently published data have shown that six weeks of endurance training markedly increased (up to four times) the gene expression and protein levels of growth factors (BDNF and GDNF) and their receptors (TrkB and Gfrα) at the low thoracic segmental level compared to untrained animals [14]. To find out whether endogenous stimulation of GFs and their receptors would affect the recovery of inhibitory neurons, we aimed to study (i) the ability of GABA-ergic (GAD 65) and glycinergic (Glyt2) neurons to express BDNF, GDNF, and their receptors after endurance training, Th9 compression, and training followed by Th9 compression in the Th10 and L2 segments, known to contain the interneurons with the greatest rhythmogenic potential [21]. In addition, we examined (ii) the impact of training on the regenerative potential of neurofilaments (NF-l), newly sprouting axons (Gap-43), and immature/mature oligodendrocytes (CNP-ase/PLP1 and APC) at the lesion site (Th9 segment) and 5 mm craniocaudally from the epicenter of injury. The functional outcome after spinal trauma was assessed (iii) using the Basso, Beattie, and Bresnahan (BBB) scoring system [22], and by measuring the transcranial motor evoked potentials (tcMEPs). We demonstrate that positive correlation between neurological outcome (assessed by BBB) and the protein level of immature oligodendrocytes (CNP-ase), mature oligodendrocytes (PLP1), and neurofilaments (NF-I) exists at the lesion site and in the caudal segment in the pretrained SCI group.

## 2. Results

### 2.1. Immunoexpression of Growth Factors in Population of GABA/Glycinergic Neurons Affected by Training and Th9 Compression

Under physiological conditions, GABA/glycinergic neurons were detected in the dorsal horns (DHs), intermediate zone (IZ), and ventral horns (VHs) of the Th10 and L2 segments. The number of GABA/glycinergic neurons in the VHs of the lumbar segment was significantly higher than in the thoracic segment (GABA-ergic, *p* < 0.0001; glycinergic, *p* < 0.01), suggesting that lumbar inhibitory neurons could control locomotor function of the hindlimbs (Figure 1). The results from double immunostaining showed colocalization of GABA^+^ and glycinergic neurons with GFs and their receptors in the ventral horn and the intermediate zone. Figure 2A–F shows the overlapping of GABA^+^ neurons with BDNF and Gfrα in the Th10 segment. A few glycinergic neurons also showed a BDNF fluorescent signal (Figure 2G–I).

We next investigated whether the immunoexpression of GFs in GABA/glycinergic neurons was enhanced by six weeks of treadmill exercise. Endurance training alone caused marked BDNF and Gfr-α expression of GABA-ergic neurons in the VHs and in deep DH layers of the thoracic segment (Figure 3A–F), and in the VHs of the lumbar segment (Figure 3G–I). Several glycine immunostained somata were colocalized with BDNF in VHs of the Th10 segment (Figure 3J–L). The results suggest that endurance training on a treadmill triggers the immunoexpression of GFs, primarily in the thoracic and lumbar GAD 65-positive neurons.

Compared to the intact control, the Th9 compression caused a decrease in the number of GABA-ergic neurons in all of the studied areas, and the number of glycinergic neurons in the VHs at the Th10 level (Table 1). Similarly, more caudally from the epicenter of the injury (e.g., in the L2 segment), a reduction of GABA and Glyt2 immunostained neurons was observed in the VHs. A significant decrease of Glyt2 neurons was also seen in the intermediate zone (Table 2). We suggest that a reduced number of inhibitory interneurons below the site of injury could result in neurogenic lower tract dysfunction, detected after SCI. The pathological mechanisms induced by Th9 compression markedly affected colocalization of GFs and their receptors with inhibitory neurons. A major decrease in such colocalization was visible in the IZ of the Th10 segment (Figure 4C,I). The lumbar area was less affected by SCI in the means of GF expression.

We observed several differences in the pretrained SCI group compared to the untrained SCI group. Table 1 shows the increase in the number of GABA-ergic neurons in the DHs of the Th10 area. Immunostaining has also shown that training + SCI substantially increased GABA- and GFRα-immunoreactive puncta in the IZ (Figure 4E,J), suggesting that preSCI training indicates the formation of new functional synaptic connections and neural circuits, resulting in the adaptive remodeling of the motor function of injured limbs after SCI. In the L2 segment, the number of Glyt2 neurons was significantly increased in the IZ (*p* < 0.001) and VHs (*p* < 0.01) (Table 2). In addition, the results show that colocalization of GFs with inhibitory GABA- and Glyt2 neurons was more marked in the pretrained than in the sedentary SCI group (Figure 4C,F,I,L).

### 2.2. Endurance Training Applied before Th9 Compression Upregulates Mature and Immature Oligodendrocytes at Lesion Site and in Its Surrounding

Oligodendrocytes are one of the main types of glial cells in the central nervous system (CNS) besides microglia and astrocytes. They are generated from oligodendrocyte progenitor cells (OPCs), following strictly orchestrated processes of migration, proliferation, and differentiation [23]. The apoptosis of oligodendrocytes begins hours after injury and continues for weeks, which is a detrimental process that contributes to the demyelination of neurons that survived the primary injury [24,25].

Considering the fact that OPCs are activated after SCI [26], we further investigated the changes in immature and mature oligodendrocytes after endurance training (endogenous trophic support), SCI, and in the pretrained SCI group. We focused exclusively on the lesion site and surrounding spinal cord tissue, since the facilitation of axonal sprouting and stimulation of neuroregeneration around the lesion site is necessary for the neural circuitry controlling these processes. As shown in Figure 5A, treadmill exercise applied 6 weeks prior to Th9 compression positively affected the oligodendrocyte (APC) population. In the T + SCI group, we observed a more pronounced APC fluorescent signal than after SCI alone. Strong APC immunolabeling was visible predominantly around the cavities near the epicenter of the injury.

Next, we evaluated the gene expression and protein levels of markers specific for immature (CNP-ase) and mature (PLP1) oligodendrocytes. Generally, after endurance training alone, the increase in gene expression and protein quantity of immature CNP-ase oligodendrocytes was seen in studied spinal cord segments (~1.5-fold and ~1.2-fold) (Figure 5B), while the mRNA and protein level of mature PLP1^+^ oligodendrocytes were not changed (Figure 5C).

As we expected, the consequences of pathophysiological mechanisms occurring 6 weeks after Th9 compression caused a marked decrease in the protein level and gene expression of both oligodendrocyte phenotypes (immature and mature) below control levels (Figure 5B,C). The most affected area was the epicenter of the injury. Next, we studied whether training before SCI could affect the CNP-ase and PLP1 levels in a craniocaudal manner. Significant differences between SCI and T + SCI groups were seen at the injury site (*p* < 0.05), where the CNP-ase and PLP1 gene expression and protein levels were significantly higher in the pretrained SCI group compared to the nontrained SCI group. Near the epicenter of the injury, we found a significant increase of immature CNP-ase expression in the cranial segment (+1, *p* < 0.05) (Figure 5B). The level of PLP1 protein was significantly higher in the caudal segment (−1, *p* < 0.05) of the T + SCI group (Figure 5C). These results indicate that endurance physical activity applied before SCI creates a suitable environment for oligodendrocyte proliferation and their differentiation needed for repairing the injured spinal cord.

### 2.3. The Upregulation of Neurofilaments and Outgrowing Axons in Spinal Cord after Endurance Training + Th9 Compression and 6 Weeks Survival

Neurofilaments are essential for radial growth stability of myelinated axons, and also for the optimal speed of electrical impulses. Loss of neurofilaments from axons severely inhibits axon radial growth, nerve conduction velocity, and the regenerative properties of axons [27,28]. Consistent with the changes of PLP1 (Figure 5C), relative gene expression of light chains of neurofilaments (NF-l) pointed out that endurance training alone did not affect NF-l gene expression in the low thoracic segments (Figure 6A). Similar results were achieved in the protein analysis of NF-l. Compared to the naive control, both NF-l mRNA expression and protein level were markedly decreased 6 weeks post-SCI. To determine whether the neurofilaments are upregulated by endurance training in the SCI model, we took advantage of the spinal cord samples collected from pretrained SCI rats. Although there was no difference in the NF-l protein level between the control and trained groups, the expression of NF-l protein in the T + SCI group was significantly enhanced (*p* < 0.001) below the lesion site when compared to the nontrained SCI group (Figure 6A).

Further, we identified changes in the gene expression and protein level of Gap-43 (marker for newly sprouted axons) through the low thoracic segments. A marked increase in Gap-43 mRNA was observed at the lesion site and in the adjacent spinal cord segments 6 weeks after SCI alone, while Western blot analysis has shown a marked increase in the Gap-43 protein level (*p* < 0.01) only at the epicenter of the injury. These data indicate a significant spontaneous restoration in the microenvironment of the thoracic spinal cord 6 weeks after its injury. Compared to the nontrained SCI group, long-term endurance training performed before SCI enhanced both the Gap-43 mRNA expression (*p* < 0.05) and the protein level (*p* < 0.01) predominantly in caudal segments, suggesting that endurance training supports GF-activated signaling pathways [14], which could promote axonal sprouting in harmed tissue (Figure 6B).

To verify the beneficial effects of pretraining, low thoracic spinal cord sections were stained with NF-l (marker for neurofilaments) and Gap-43 antibodies. As shown in Figure 6C, NF-l-positive fibers in the T + SCI group were more markedly distributed through the dorsolateral funiculi of the Th10 segment compared to the nontrained SCI group. In pretrained animals, we also noticed a more pronounced Gap-43 fluorescent signal and longer fibers overlapping with NF-l and Gap-43 markers (white rectangles; Figure 6C).

### 2.4. Endurance Training before SCI Improves Micturition, BBB Score, and Transcranial Motor Evoked Potentials

All experimental animals in the SCI and T + SCI groups were regularly monitored. We assessed the time-dependent appearance of spontaneous micturition, neurological BBB score, and transcranial motor evoked potentials (tcMEPs). After SCI, the bladder was manually expressed 2–3 times daily until spontaneous micturition was achieved. We noted that the spontaneous voiding started earlier in the T + SCI group (13.6 ± 3.37 days) than in the SCI group alone (16.44 ± 1.59 days) (*p* = 0.034). It seems that voluntary urination occurs with better time in the pretrained group than in the group with SCI, due to better neurogenic function and neurotrophic environment around the lesion site.

One day post-SCI, all the animals suffered from bilateral hindlimb paralysis (BBB score = 0–1; SD ± 0.57) (Figure 7). During the first 6 days after Th9 compression, both studied groups (SCI and T + SCI) achieved approximately similar neurological scores in BBB testing. Slight differences in hindlimb locomotor function recovery appeared from day 12, reaching the mean score of 5.75 (SD ± 1.41) in the SCI group and 7.44 (SD ± 1.21) in the T + SCI group. Significant improvement in the motor activity was noticed only at the end of survival (42 days post-SCI); the mean score for functional outcome was graded 12.06 ± 0.92 points in the T + SCI group and 10.58 ± 0.97 points in the SCI group.

The tcMEPs response to subcutaneous cerebral cortex stimulation performed before spinal compression and in 2-week intervals after trauma showed that endurance training had a positive effect on the recorded tcMEPs peak latency (ms). Animals, after endurance training and Th9 compression, had a shorter response to the stimulus compared to the nontrained SCI animals. After the first post-SCI measurement (2 weeks), we noted significant differences between the SCI (9.59 ± 1.4) and T + SCI groups (7.48 ± 0.43). In pretrained animals, the response to stimulation was significantly shorter (*p* < 0.01). At the end of survival (6 weeks post-SCI), the latency of tcMEPs was significantly longer in the SCI group (8.59 ± 0.7) compared to pretrained animals (7.14 ± 0.57) (Figure 8). These results indicate that long-term physical activity applied before traumatic SCI affects the tcMEPs recovery and could improve motor function.

### 2.5. Correlation between BBB Neurological Score, Protein Levels, and Gene Expression for Axonal and Glial Markers

Pearson’s correlation analysis was performed to identify the relationship between the protein levels and/or the gene expression of markers specific for immature and mature oligodendrocytes (CNP-ase, PLP1), neurofilaments (NF-l), newly outgrowing axons (Gap-43), and BBB neurological score. Our results show that there is a significant negative correlation (*p* < 0.05) between the protein level for CNP-ase and hindlimb motor recovery in the nontrained SCI group at the epicenter of injury and cranially (Figure 9A). We also found a negative correlation between the PLP1 protein level and the behavioral outcome at the lesion site (r = −0.552, *p* = 0.335) (Figure 9C), and between the expression of CNP-ase and/or PLP1 and the functional recovery just below the site of injury and/or at the lesion site (Figure 10A,C).

Data from the T + SCI group indicate that the endurance training before spinal cord trauma affected the correlation between the levels of immature CNP-ase oligodendrocytes and the recovery of motor activity. A positive correlation was found only in cranial (r = 0.505, *p* = 0.386) and caudal (r = 0.525, *p* = 0.364) segments (Figure 9B). The correlation coefficients for mature PLP1 protein level and/or PLP1 gene expression and locomotor recovery were positive caudally (r = 0.646, *p* = 0.240) (Figure 9D) and at the lesion site (r = 0.496, *p* = 0.504) (Figure 10D). The results indicate that long-term endurance training applied before SCI positively affects the correlation between the protein level of both oligodendrocyte phenotypes and functional outcome predominantly in segments below the site injury.

Data from the Th9 compression group (SCI) show a negative correlation between axonal markers (NF-l and Gap-43) and BBB score in the whole craniocaudal manner (Figure 11A,C and Figure 12A,C), except for a positive correlation (r = 0.536, *p* = 0.464) in NF-l expression below the site of injury (Figure 12A). In the T + SCI group, the functional recovery highly correlated with the protein levels for NF-l at the lesion site (r = 0.539 (*p* = 0.349) and caudally (r = 0.499, *p* = 0.392) (Figure 11B). The measurements of the NF-l gene expression correlated with the neurological score at the epicenter of injury (r = 0.680, *p* = 0.321), while the correlation in T + SCI in the caudal segment was lower (r = 0.313, *p* = 0.687) than in the SCI group (r = 0.536, *p* = 0.464) (Figure 12A,B). Overall, analysis of an NF-l axonal marker, but not the Gap-43 marker, revealed that long-term endurance training applied before SCI positively correlates with better recovery.

## 3. Discussion

### 3.1. Modulation of Neurotrophic Support in Inhibition Neurons after Endurance Training, SCI, and T + SCI

Previous studies clearly demonstrated that adaptive changes/plasticity in some motor-related CNS mechanisms [29,30] could play a major role in training-induced recovery. Experimentally conducted endurance training increased BDNF and TrkB expression in the locomotor-related brain areas (striatum, cerebellum) and plasma [30], markedly upregulated (up to four times) mRNA expression and protein levels of BDNF, GDNF, and their TrkB and Gfrα receptors in the low thoracic segments (Th8–Th10) [14], increased immunoreactivity of BDNF, NT-4, and TrkB in L3 and L4 lumbar spinal cord segments [31], as well as upregulated BDNF levels in skeletal muscle [32]. In the present study, 6 weeks of endurance training caused the rearrangement in neurotrophic regulation of inhibitory GABA/glycinergic neurons at the Th10 and L2 levels, which likely are essential in a functionally meaningful manner.

It is well known that the basic rhythm and walking patterns are generated by component neurons of a spinal neural network, known as the locomotor central pattern generator (CPG) [33,34]. While these neurons are distributed across numerous spinal cord segments, the interneurons with the greatest rhythmogenic potential, are located in the laminae VII, VIII, and X of the lower thoracic and upper lumbar spinal cord [21]. After training, we found a colocalization of BDNF and Gfrα with GABA-ergic neurons in areas known to contain primary afferent terminals from the hindlimb muscles (i.e., the medial and lateral regions of laminas IV–VI) [35], the VHs of the Th10 segment, and in the VHs of the upper lumbar L2 segment. However, only some Glyt2-IR cell bodies were also stained with BDNF/Gfrα in the VHs of the Th10 segment.

Th9 compression considerably reduced the number of inhibitory GABA/glycinergic neurons in both Th10 and L2 segments. Colocalization of GFs with GABA immunostained neurons was reduced predominantly in the IZ, while the L2 segment was less affected. Based on classical descriptions, the subpopulation of interneurons in the IZ is composed of multiple cell types, including Renshaw cells and Ia inhibitory interneurons. These interneurons diverge in their physiological properties because they display neuronal input specificity (e.g., from sensory vs. motor neurons), and form inhibitory microcircuits that are tailored to individual limb muscles. It has been shown that these interneurons play a role in locomotion because they are able to alter the speed of the step cycle, and a decrease in their number after SCI could result in prolonged activation of spinal motoneurons [36,37]. These motoneurons innervate flexor muscles, disrupting the stereotypical flexor–extensor phases of contraction during locomotion [38]. Since a decrease of GABA/glycinergic neurons in specific spinal cord regions just below the lesion site is consistent with a decrease of neurotrophic support, we suggest that adaptive changes in Th10 and L2 segments could modulate the spinal circuits involved in the control of posture and locomotion.

Recently, Li et al. [39] revealed that 4 weeks of exercise training increased the levels of BDNF, TrkB, GAD 65/67, and KCC2 in the distal region of the rat spinal cord after Th10 contusion, and that the blocking of TrkB signaling decreased the effects of exercise training. In fact, these results indicate that exercise training increases GAD 65/67 and KCC2 expression in the spinal cord via TrkB signaling and could improve motoneuronal hyperexcitability and spasticity induced by incomplete SCI. Novel data from our lab also showed that 6 weeks of treadmill training prior to SCI markedly modulated the PLCγ/Ras/Erk1/2, PI3k/Akt, PKC, and PLCγ-CAMKII signaling pathways via BDNF-TrkB signaling cascade just around the lesion site [14]. The activation of these signaling pathways plays a key role in the neurological outcome of pretrained SCI animals. The present data show that pretraining modulates the GF-induced inhibitory capacity of neurons in specific spinal cord regions. The ability of pretrained SCI rats to perform frequent and consistent weight-supported plantar steps, as well as occasional front–hindlimb coordination 6 weeks post-SCI, could be a result of training-regulated synthesis of GABA/glycine in the deep DH neurons and IZ of the T10 segment, and glycine in the neurons of the IZ and VHs of L2 segments, which in turn could modulate their excitation in the appropriate motor pools. We suggest that GF-induced regulation of GABA/glycine synthesis in Th10 and L2 segments affects neuronal systems that directly contribute to the execution of a given motor task. Since the production of locomotion also requires supraspinal and sensory inputs originating outside the spinal cord [18,40,41,42], to explain how regular exercise before SCI could modulate networks of inhibitory interneurons, leading to better functional outcome, requires more precise morphological and network connectivity analyses.

Over the past decade, remarkable progress has been made in neural stem cell therapy for SCI [43]. Su et al. [44] examined the reprogramming of endogenous scar-forming astrocytes into neurons in the adult mouse spinal cord and found that a high-mobility group of DNA-binding domain transcription factor, SOX2-induced adult neurogenesis can generate mature neurons with features of GABA-ergic interneurons in injured valproic acid (VPA)-treated spinal cords. Although the number of converted neurons was low, the authors have shown that new neurons were capable of forming synapses with pre-existing CHAT + motoneurons, suggesting potential integration into the local neural network of the injured spinal cord [44]. Currently, cell therapy is one of the effective therapeutic strategies for SCI; however, more studies are required to verify the potential of such treatment to regenerate and/or remyelinate axons, and to incorporate new neural cells into the milieu of a traumatic SCI [43].

### 3.2. Physical Activity Applied before SCI Induced Glial and Axonal Restoration

Pathogenesis of the SCI indicates poor axonal regeneration that leads to irreversible loss of neuronal connectivity and severe functional impairment [45,46]. In addition to axonal damage, traumatic injury affects the whole cell population, including glial cells providing an essential neuronal support. Oligodendrocytes are glial cells which have unique properties in the nervous system. They play a crucial role in the maintaining of homeostasis, physiology, and neural plasticity. Additionally, their products may act in favor of spinal cord repair after injuries [47].

Our data showed that spinal compression at the low thoracic level caused death of CNP-ase, PLP1, and APC oligodendrocytes, and massively affected the NF-l- and Gap-43-positive axons at the epicenter of the injury (5 mm) and in nearby segments (5 mm cranially and caudally). The largest loss of oligodendrocytes and NF-l was found at the lesion site. We also demonstrated severe paralysis of the hindlimbs after Th9 compression. The BBB scores showed a lack of movement in both hindlimbs for more than 3 weeks after trauma, and tcMEPs measurements clearly indicated the longest latency and smaller amplitude of the measured signals compared to control. These data indicate that the recovery of oligodendrocytes (as the primary source of myelin production) and neurofilaments (essential for maintaining axonal caliber and the transmission of electrical impulses along axons) at the lesion site and in the cranial and caudal segments immediately adjacent to the lesion could be essential for modulation of the key spinal microcircuits, leading to a better outcome.

One of the crucial mechanisms in the regenerative process after spinal cord trauma is axonal remyelination. Remyelination of the myelin sheaths in the injured axons has been shown to improve axonal conduction, and thus functional recovery [48]. During development, and also after injuries, several cytokines and/or growth factors act as inhibitors/activators of oligodendrocyte lineage development and maturation. Considering the fact that a training-induced increase of GFs activates intracellular signaling pathways [14], we looked into the repair of immature and mature oligodendrocytes. A deeper analysis showed that endurance training alone increased the gene expression and protein level of CNP-ase oligodendrocytes (~16%) at the low thoracic level, but did not alter the expression/protein level of PLP1 oligodendrocytes. In the pretrained SCI group, however, the protein level of both immature and mature oligodendrocytes was increased by 13% at the lesion site and just below the injury. This likely reflects both the neuroprotective properties of the training, as it maintains increased expression of GFs in the spinal cord and increased expression of PI3k, PDK1, and Akt genes (responsible for cell survival), and upregulation of RAS, ERK1, and ERK2 genes playing an essential role in differentiation, cell proliferation, and axonal outgrowing [14]; and in addition, the activation of neuroprotective markers which are required for spontaneous post-SCI recovery [49].

Since BDNF plays a key role in the promotion of oligodendrocyte survival, differentiation, and maturation [48], we assume that BDNF, via its high-affinity receptor TrkB, activates the signaling transduction through PI3K/AKT and Erk1/2, and thus affects the differentiation and maturation of oligodendrocytes into the myelinating phenotype. It seems that enhanced activity of GF-stimulated signaling pathways responsible for cell survival and regeneration could improve the neural connectivity in the damaged tissue and affect the motor and sensory functions. Recently, Siebert and Osterhout [50] indicated that neurotrophic factors, especially BDNF, GDNF, and NT-3, promote axonal regeneration after spinal trauma and have a profound effect on oligodendrocyte lineage cells. Their results revealed that treatment with growth factors in vitro can initiate the process of OPC differentiation in the detrimental environment of an injured spinal cord rich in chondroitin sulfate proteoglycans (CSPGs). Similarly, as with BDNF, a cytokine membrane-bound neuregulin (Nrg1) could play an essential role in differentiation and maturation of oligodendrocytes. It acts through their ErbB receptors, and regulates the expression of myelin-related genes via Jak/STAT, Erk/MAPK, and PI3/AKT that are associated with cell survival, proliferation, and differentiation. Nrg1 enhances the differentiation of spinal stem cells into oligodendrocytes, while decreasing astrocyte differentiation [48,51]. Ding et al. [52] studied the effect of Nrg1 on the oligodendrocyte lineage cells. They found that Nrg1, through PI3K/AKT/m-TOR signaling, converts reactive astrocytes toward oligodendrocytes and increases the protein levels of myelin basic protein, neurofilaments, and the immunoexpression of CNP-ase. They also observed a significant improvement of neurological BBB scores in the Nrg1-treated group. Collectively, these findings offer a fresh insight, that certain neurotrophins and/or Nrg1 could stimulate the signaling pathways responsible for oligodendrocyte-dependent axonal remyelination and functional recovery after SCI, and could also neutralize the inhibitory effects of CSPGs at the lesion site. The loss of motor, sensory, and autonomous functions after spinal trauma is mainly caused by disrupted neural circuits and severed axons [53]. For years, regeneration of axons and formation of new synapses after SCI has been a highly challenging problem [54,55]. Our study has shown that upregulation of Gap-43 and NF-l after endurance pretraining of SCI rats (the findings supported by immunohistochemical, protein, and gene expression analyses) is clearly visible below the injury site, where both neurofilaments and newly sprouting axons show great potential to overcome the inhibitory obstacles in the injured tissue. We suggest that a BDNF- and GDNF-rich environment overcame the detrimental mechanisms occurring at and around the injury epicenter. The importance of broad axonal regeneration and sprouting has been shown in a study by Alibardi [56]. The author noticed that after lumbar spinal cord transection in lizards (*Podarcis muralis*), numerous intensely labeled axons (NF200) were seen 19–22 days post-injury. The number of immunostained axons increased dramatically 3 months after SCI. Only a few Gap-43-positive newly sprouting axons regenerated through the transected area; however, they contributed to the reactivation of limited recovery of the hindlimbs that occurred 1 month after injury. At this period of time, lizards could move their hindlimbs for walking and running, but they could not lift up their bodies using the hindlimbs when leaning on a wall. However, reptilian regeneration appears to be slightly better than in mammalian models (e.g., rats, mice), but the limitations are still profound compared to anamniotes. Recently, the relationship between NF-l and Gap-43 and function outcome after Th9 compression was also confirmed by applying an oscillating field (OF) stimulator (50 µA) [57]. An OF stimulator reverses the polarity of the electric field in 15 min intervals to initiate the simultaneous regeneration of both ascendent and descendent neuronal pathways. The highlights of this study proved that OF stimulation can effectively improve neurofilament restoration in the thoracic spinal cord at the injury epicenter, as well as at the adjacent segments. Behavioral analyses supported the hypothesis that axonal regeneration has a massive impact on functional recovery. The results showed significant improvement in BBB neurological scores and in hot plate tests that reflect the regeneration of both motor and sensory fibers. A study by Jing et al. [58] also proved that SCI treatment with OF stimulation improved remyelination after spinal trauma, which resulted in BBB score improvement from the 4th to 8th weeks post-trauma. They also found that OF stimulation significantly reduced the latency of tcMEPs after treatment through the whole survival period. In addition, Weishaupt et al. [59] reported that electrical stimulation increased the excitability of the spinal cord neural network, specifically caudally from the lesion site. It promoted axonal outgrowth and increased the gene expression of BDNF and its receptor TrkB, probably via the extracellular signal-regulated kinase (ERK) pathway. In our study, we also pointed out that BDNF (as a result of pretraining in the SCI model), among its many actions, has been shown to stimulate axonal restoration, as well as immature and mature oligodendrocyte functions, probably through specific growth factor-dependent signaling pathways. The fundamental question is how to incorporate our findings to clinical practice. The most important challenges that have to be considered in clinical use of neurotrophins in SCI patients are (a) very short half-life, (b) in most cases they do not cross the blood–spinal barrier, (c) invasive techniques for their administration, (d) adverse side effects, and (e) possible nonspecific effects on uninjured tissue [7]. Currently, there is no treatment protocol for patients with SCI. There are only preclinical trials that have been performed in experimental models. The findings of our study show that repair after SCI is better in individuals who performed regular endurance training, and what is more important, we partially provide the evidence of mechanisms responsible for increased regeneration of axons and glial cells in the cranio–caudal extent of the injured spinal cord. Most of our understanding of the regeneration after SCI derives from experimental studies performed on well-characterized SCI models, which do not precisely reflect the variability of human SCI pathology. Recent studies addressing differences between athletes and nonathletes with SCI focus mainly on the mental health and overall quality of life [60,61]. The results indicate that people participating in sport activities appropriate to their degree of disability have a significantly higher quality of life and life satisfaction compared to people not involved in any sport.

Precise assessment of motor deficits after SCI is crucial for understanding the mechanisms of functional outcome and testing after any therapeutic interventions. In the present study, the favorable effects of endurance training were detected by behavioral BBB testing and measuring the tcMEPs. Significant differences between SCI and pretrained SCI groups were seen during the 6-week period of survival. Two weeks post-SCI, improvement in the response to cerebral cortex stimulation and BBB score was clearly visible in the T + SCI group. A similar pattern of greater recovery was also noticed at the end of survival (42 days post-SCI). We recently pointed out that astrocytes of the dorsolateral area are mainly responsible for GF expression after endurance training [14]. Since it is known that the spinal cord neurons and glia die at the lesion site (2–4 mm) within minutes/hours after SCI, and cell types, including neurons, astrocytes, microglia, and oligodendrocytes surrounding the lesion site are lost in a delayed manner [62,63], we suggest that training-potentiated activation of GFs has an impact on restoration of various cell populations interacting with each other within the lesion site and in surrounding spinal cord tissue. We assume that mechanisms controlling proliferation and cellular identity after endogenously stimulated regeneration, together with the testing of the safety and efficacy of delivery systems of neuroprotective agents and cells in combination with long lasting rehabilitation and/or OS stimulation, could regain motor and sensory function after SCI. In addition, understanding the role of myelin basic protein, which maintains the correct structure of myelin, interacting with the lipids in the myelin membrane, could be the key to uncovering the mechanisms underlying repair events in this experimental model.

## 4. Materials and Methods

### 4.1. Animals and Experimental Design

Adult female Wistar rats (230–280 g weight, 3–4 months of age) were used for the experiment. The rats were randomly divided into four experimental groups: (1) intact control (*n* = 13); (2) six weeks endurance training on a treadmill (*n* = 13); (3) Th9 compression (40 g/15 min) (*n* = 13); (4) treadmill training followed by Th9 compression (*n* = 13). After SCI, the rats were housed individually in standard cages with a lid to hold a water bottle and food for ad libitum eating and drinking in a controlled environment with a temperature of 22–24 °C and humidity at 45–50%. After surgical procedure, the rats survived six weeks.

### 4.2. Endurance Training

The experimental animals in group 2 (6 weeks endurance training) and group 4 (pretraining and Th9 compression) were scheduled for endurance training on a rodent treadmill (Treadmill LE 8710; Bioseb, Pinellas Park, FL, USA) 5 days per week for 6 weeks. The training session always started on Monday for 40 min/day, and the duration was gradually increased by 5 min every day until Friday. The intensity of the training was also gradually increased. During the first week, the training speed was set to 16.2 m/min, the second week to 19.8 m/min, the third week to 22.2 m/min, the fourth week to 24.6 m/min, and the fifth and sixth weeks were set to 27.6 m/min.

### 4.3. Surgical Procedure

The animals in group 3 (Th9 compression) and group 4 (pretraining and Th9 compression) were, prior to surgery, gently weighed and anesthetized with 2–4% isoflurane (Vetpharma, Barcelona, Spain) 1.5–2.0 L/min oxygen mixture, delivered by facemask. After disinfection of the surgery site with Betadine (EGIS Pharmaceuticals PLC, Budapest, Hungary), a midline incision was made and the Th9 vertebra was carefully exposed. The backbone was fixed by clamping with surgical forceps in a stereotaxic frame. Spinal cord compression was induced using a compression device with a plastic impactor with a weight of 40 g. After 15 min of compression, the impactor was removed and the wound sutured with sterile silk (USP 4/0, Chirana, Prague, Czech Republic). A stable core body temperature was maintained at 37.0 °C during the surgical procedure by a homeothermic heating pad (ATC1000), WPI, Sarasota, FL, USA). After Th9 compression, the rats were housed individually in standard cages. Postoperative care began immediately after surgery; animals received Amoksiklav antibiotic (Sandoz Pharmaceuticals, Ljubljana, Slovenia; 30 mg/kg IM) and Novasul analgesic (Richterpharma, Wels, Austria; 2 mL/kg IM) for three consecutive days. Sterile saline solution (5 mL) (Bieffe Medital S.P.A., Grosotto, Italy) was subcutaneously injected to prevent dehydration after surgical procedure. After Th9 compression, rats suffered from bladder areflexia, so until spontaneous voiding was restored, animals’ bladders were manually expressed twice a day. At the end of survival, animals were sacrificed either by means of decapitation (for RT-PCR and WB analyses) or transcardial perfusion (for immunohistochemical analysis).

### 4.4. Gene Expression Analysis (RT-PCR)

After decapitation, the spinal cords (*n* = 4 per group) were removed from the spinal canal and subsequently dissected into three 5 mm segments (the lesion site, cranial, and caudal segments). Samples were carefully frozen with liquid nitrogen and stored at −70 °C until further processing. Spinal cord tissues were homogenized and total RNA was isolated using Trizol reagent (Thermo Fisher Scientific, Waltham, MA, USA) according to the manufacturer’s protocol. The concentration and quality of RNA were checked using spectrophotometric analysis, measuring the 260/280 ratio (NanoDrop2000c, Thermo Fisher Scientific, Waltham, MA, USA). Using a high-capacity cDNA reverse transcription kit (AB Applied Biosystems by Thermo Fisher Scientific, Waltham, MA, USA), cDNA was synthesized from 2000 ng of total RNA using a T1000™ Thermal Cycler (Bio-Rad, Hercules, CA, USA). The amplification of cDNA (10 ng per reaction) was performed using the CFX96TM Real-Time System (Bio-Rad, USA) and the primers (Table 3) were designed with Geneious software (Biomatters, Ltd., Auckland, New Zealand). Power SYBR Green PCR MasterMix (Applied Bioscience by Thermo Fisher Scientific, USA) was used with 1 μL of each primer. The amplifications were run routinely in duplicates under the following conditions: 10 min at 95 °C, followed by 50 cycles consisting of 15 s of denaturation at 95 °C, and annealing/extension for 1 min at 60 °C. Relative gene expression of the analyzed targets was normalized using the 18S RNA reference gene. The results were calculated using the ΔΔCt method.

### 4.5. Western Blot Analysis

Tissue samples collected from each experimental group (*n* = 5 per group) were homogenized by a Minilys homogenizer (Bertin Instruments, Montigny-le-Bretonneux, France) in extraction buffer (RIPA, Sigma-Aldrich, St. Louis, MO, USA) with protease inhibitor (Roche Diagnostics, Rotkreuz, Switzerland). The protein concentration of supernatants collected after centrifugation (25 min at 14,000 rpm) was quantified with a Pierce™ BCA Protein Assay Kit (Thermo Fisher Scientific, USA). An equal concentration of 20 µg protein per well was loaded and separated using SDS-PAGE (12%; 90 min/100 V) and transferred (30 min/15 V) onto a PVDF membrane (Bio-Rad, USA) using Trans-blot Turbo transfer system (Bio-Rad, USA). Next, the membranes were blocked for 1.5 h in 5% nonfat milk in Tris-buffered saline/Tween^®^20 (TBS-T) solution at room temperature. Blocking was immediately followed by overnight incubation (at 4 °C) with primary antibodies (anti-CNP-ase, 1:500, Cell Signaling, USA; anti-PLP1, 1:500, Cell Signaling, USA; anti-NF-L, 1:1000, Cell Signaling, USA; anti-Gap-43, 1:500, Millipore, Darmstadt, Germany). Blots were washed four times with TBS-T buffer and incubated with anti-rabbit IgG HRP conjugated secondary antibody (1:5000; Santa Cruz Biotechnology, Dallas, TX, USA) for 1.5 h. Following four washes with TBS-T buffer, protein bands were visualized using the enhanced chemiluminescence substrate (ECL, Thermo Fisher Scientific, USA) and scanned with the Fusion FX Imaging system (Vilber, Marne-la-Vallée, France). Afterward, membranes were incubated in Restore™ Plus Blot Stripping Buffer (Thermo Fisher Scientific, USA) for 10 min and re-used for incubation with anti-β-actin HRP antibody (1:20,000; Abcam, Cambridge, UK) for 1.5 h. The optical density of the bands was quantified using Quantity One 4.6 Software (Bio-Rad, USA). The levels of measured proteins were estimated as the ratio of the optical density of the examined protein to β-actin, which served as a normalization protein.

### 4.6. Immunohistochemical Analysis

Spinal tissue (Th8, Th9, Th10, and L2 segments, *n* = 3 per group) obtained after transcardial perfusion with saline solution (300 mL) and 4% paraformaldehyde (300 mL) was cryoprotected (30% sucrose) and sectioned in a Leica CM1850 cryostat to 25 µm thick transverse and longitudinal sections. Selected sections were washed four times in PBS with 0.3% Triton X-100 (PBS-T; Sigma-Aldrich, St. Louis, MO, USA) and blocked for 30 min in 5% normal goat serum in PBS-T. Subsequently, 10 sections from each 5 mm segment (every 20th cut) were incubated with the following primary antibodies overnight at 4 °C: mouse anti-BDNF (1:150; Abcam, Cambridge, UK), rabbit anti-GDNF (1:50; Santa Cruz, Dallas, TX, USA), mouse anti-Gfrα (1:50; Santa Cruz, USA), rabbit anti-GABA-GAD 65 (1:100; Abcam, Cambridge, UK), rabbit anti-Glyt2 (1:600; Thermo Fisher Scientific, USA), mouse anti-NeuN (1:1000; Abcam, Cambridge, UK), mouse anti-APC-ase (1.200; Millipore, Darmstadt, Germany), rabbit anti-NF-l (1:100; Cell Signaling, Danvers, MA, USA), and mouse anti-Gap-43 (1:500; Cell Signaling, Danvers, MA, USA). Labeled sections were then washed four times in PBS-T and incubated with FITC goat anti-mouse and FITC goat anti-rabbit IgG secondary antibodies (1:200; Jackson Immunoresearch, West Grove, PA, USA) for 2 h at room temperature. After incubation, spinal sections were washed with 0.3% PBS-T, additionally stained with DAPI (Roche, Mannheim, Germany), washed again, and rinsed in distilled water. Fluoromount mounting medium (Serva, Heidelberg, Germany) was then used to adhere the cover slips to the microscope slides. Digital images of the stained sections were captured using an Olympus BX51 (Tokyo, Japan) fluorescent microscope and acquired using QuickPHOTO MICRO 2.3 software (Promicra, Prague, Czech Republic) All obtained photographs were captured at the same exposure time (1/9 s) with 0 corrections. The resolution of the microphotographs was set to 1360 × 1024 pixels.

### 4.7. GABA/Glycinergic Cell Quantification

GAD 65- and Glyt2-positive cells were counted using Image J software (NIH, Bethesda, MD, USA) in the selected rectangular areas (1.1 mm × 1 mm) of the intermediate zone, dorsal, and ventral horns (see Figure 13). Cell detection was performed from ten randomly selected transverse spinal sections caudally from the lesion site (Th10 and L2). Digitalized images of the sections used for the quantification were converted into binary images, and the number of GAD 65- and Glyt2-positive cells were counted using the “analyze particles” tool.

### 4.8. BBB Locomotor Testing

The recovery of the motor function of paralyzed hindlimbs was evaluated using the BBB locomotor scale method (*n* = 9 per group), a standardized locomotor rating test (Basso et al., 1995). The scale ranges from zero points (complete paralysis) to 21 points (complete physiological mobility). The BBB scale reflects a combination of joint movement, stepping, forelimb and hindlimb coordination, trunk position and stability, tail position, and paw placement. Behavioral testing was performed in an open field once every six days during survival by two evaluators.

### 4.9. Transcranial Motor Evoked Potentials (tcMEPs)

Prior to measurements, a dose of Ketamine 50 mg/kg (Richter Pharma AG, Wels, Austria) was administered intraperitoneally to rats for induction of anesthesia. The head region of the rats (*n* = 4 per group) was shaved and disinfected with Betadine. Transcranial motor-evoked potential (tcMEPs) stimuli were applied with an isolated current stimulator (Digitimer, Welwyn Garden City, UK) through two needle electrodes placed subcutaneously above the sensorimotor cortex. Motor evoked potentials were recorded from the *musculus gastrocnemius* using two needle electrodes located 1 cm from each other. A reference electrode was placed subdermally on the contralateral side of the animal’s trunk. The duration of each stimulation pulse was set to 1 ms, with an amplitude of 40 mA. The interval between successive measurements was 20 s and values were averaged from 10 to 15 individual responses. The signal was then amplified and filtered by recording amplifier (ADInstruments, Colorado Springs, CO, USA), with high-pass and low-pass filters set to 100 Hz and 10 kHz, respectively. The latency and amplitude of tcMEPs and waveforms were recorded using LabChart software (ADInstruments, USA) and stored for further analysis. tcMEPs recording was carried out in each experimental group prior to SCI and then in 2-week intervals until the end of survival.

### 4.10. Statistical Analysis

The data obtained from the RT-PCR, Western blot, and immunohistochemistry were analyzed using a one-way analysis of variance (ANOVA) and post hoc Tukey’s HSD test. Student’s parametric *t*-test was used to analyze the data from the measurements of transcranial motor evoked potentials, and the repeated measures of variance (repeated one-way ANOVA—multiple comparisons) were used to conduct statistical analysis for behavioral testing (BBB scores). Correlation analyses between BBB neurological sores and the markers of axonal regeneration (CNP-ase, PLP1, NF-I, Gap-43) were performed with the Pearson correlation test in a group of animals (*n* = 9) chosen for both the Western blot (*n* = 5) and RT-PCR analyses (*n* = 4). All the data were analyzed using Graph Pad Prism version 8.01 (USA) and the significance level was set as *p*-Value less than 0.05 (*).

## 5. Conclusions

Taken together, we show that long-term physical activity applied before spinal cord compression had a beneficial effect on the microenvironment within the lesion site and in the surrounding spinal cord tissue at several levels. Firstly, six weeks of training increased the gene expression and protein level of immature CNP-ase oligodendrocytes at the Th10 level and caused rearrangements in neurotrophic regulation of inhibitory GABA/glycinergic neurons at the Th10 and L2 levels containing interneurons with rhythmogenic potential. Secondly, training + SCI positively affected specific markers for immature and mature (CNP-ase, PLP1) oligodendrocytes at the lesion site and caudally, and increased the number of GABA- and glycine-ergic neurons in specific spinal cord regions (dorsal horn, intermediate zone, and ventral horn). Thirdly, the protein levels of CNP-ase, PLP1, and NF-l, but not Gap-43, positively correlated with the neurological outcome just below the site of injury.

## Figures and Tables

**Figure 1 ijms-24-08616-f001:**
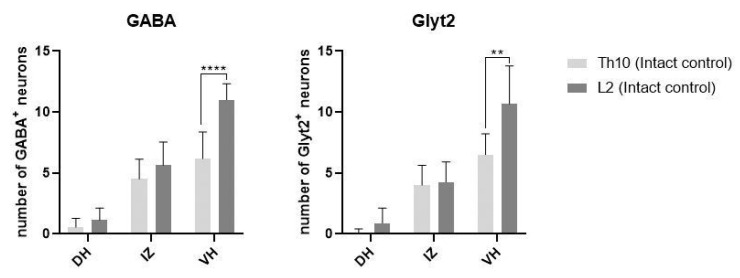
Physiological intersegmental differences between the number of GABA/glycinergic neurons in Th10 and L2 segments in dorsal horns (DHs), intermediate zone (IZ), and ventral horns (VHs). Results are presented as mean ± SD (*n* = 3). Data were statistically evaluated using Student’s parametric t-test; ** *p* < 0.01; **** *p* < 0.0001.

**Figure 2 ijms-24-08616-f002:**
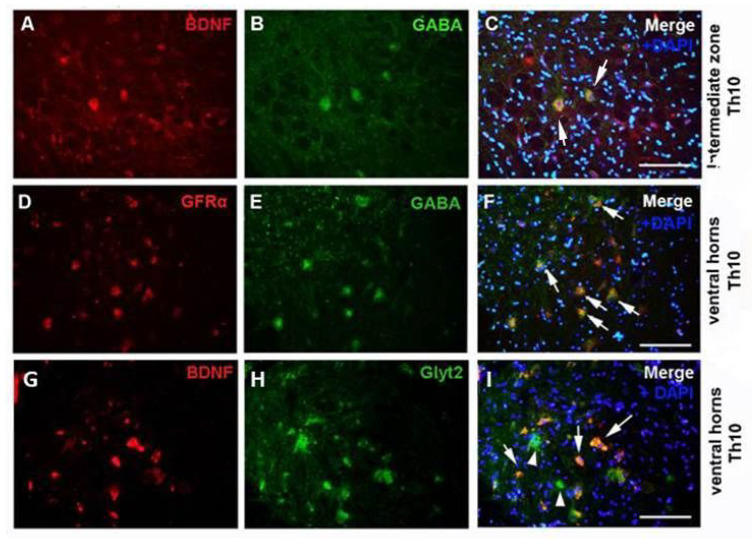
Immunohistochemical analysis of GABA (**B**,**C**,**E**,**F**; green) and Glyt2 (**H**,**I**; green) neurons with BDNF (**A**,**C**,**G**,**I**; red) and GFRα (**D**,**F**; red) in the intermediate zone and ventral horns of Th10 spinal segments in naïve control group. White arrows show the overlapping of the inhibitory neurons with BDNF or GFRα and DAPI, heads show Glyt2 neurons (*n* = 3). Scale bars: (**A**–**I**), 100 µm.

**Figure 3 ijms-24-08616-f003:**
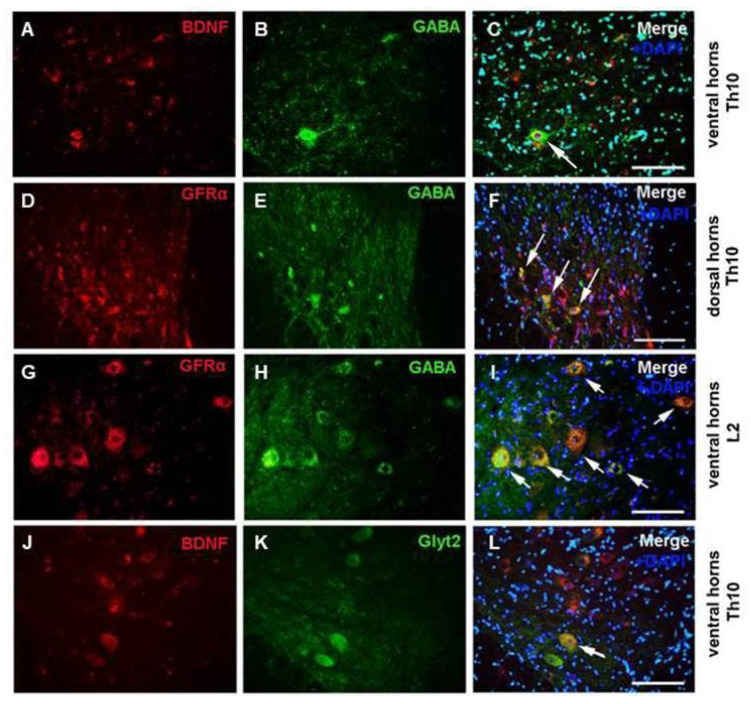
Fluorescent labeling of BDNF growth factor **(A**,**C**,**J**,**L**; red) and GFRα receptor (**D**,**F**,**G**,**I**; red) after six weeks of endurance training in specific areas of Th10 and L2 segments of spinal cord. Microphotographs also show the overlapping of these factors with GABA (**C**,**D**,**I**; green) and Glyt2 (**L**; green) inhibitory neurons (white arrows) (*n* = 3). Scale bars: (**A**–**L**), 100 µm.

**Figure 4 ijms-24-08616-f004:**
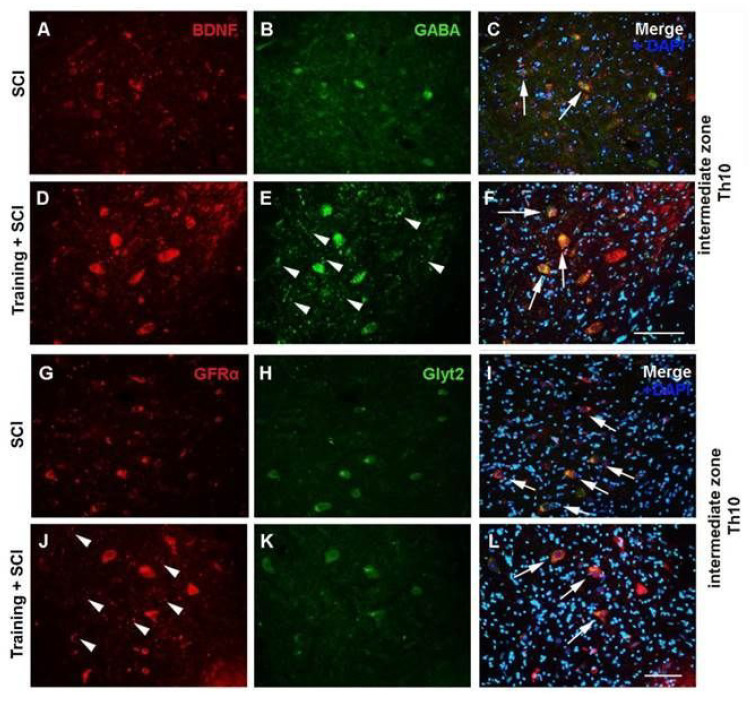
Immunoexpression of BDNF (**A**,**C**,**D**,**F**; red), GFRα (**G**,**I**,**J**,**L**; red), GABA (**B**,**C**,**E**,**F**; green) and Glyt2 (**H**,**I**,**K**,**L**; green) neurons in nontrained (SCI) and pretrained (T + SCI) groups with Th9 compression. Differences in the experimental groups are shown in the intermediate zone of Th10 segment (just caudally from the lesion site). White arrows indicate the overlapping of BDNF with GABA, GFRα with Glyt2 and DAPI, heads show GABA- and GFRα-immunoreactive puncta (*n* = 3). Scale bars: (**A**–**L**), 100 µm.

**Figure 5 ijms-24-08616-f005:**
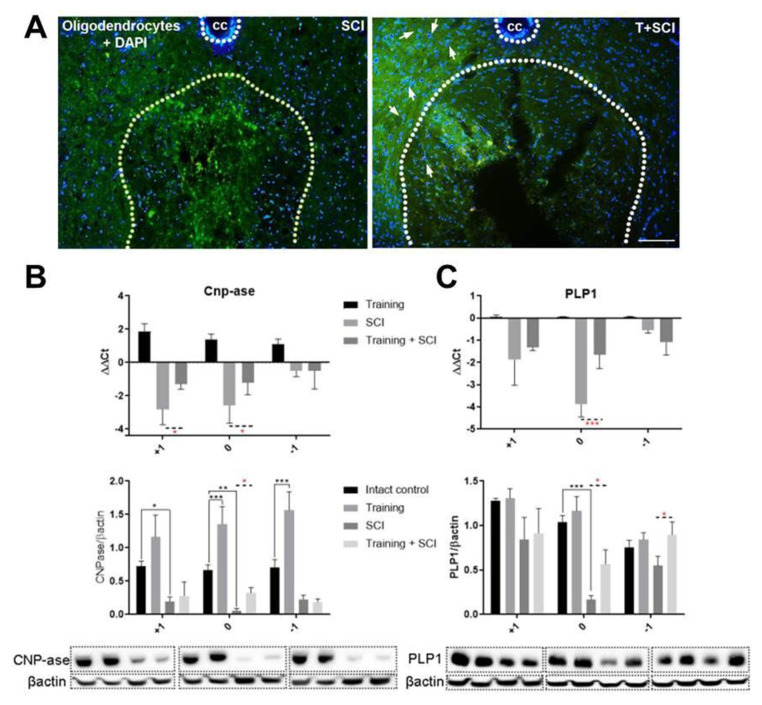
Immunohistochemical, relative gene expression, and protein level analyses of oligodendrocytes in thoracic spinal cord. Fluorescent APC staining of mature oligodendrocytes colocalized with DAPI in SCI and T + SCI groups in caudal segment (−1) (*n* = 3) (**A**). Levels of ΔΔCt for immature CNP-ase oligodendrocytes and the ratio of CNP-ase/βactin indicate the effect of endurance training on intact vs. traumatized tissue (**B**). Levels of ΔΔCt for mature PLP1 oligodendrocytes and the ratio of PLP1/βactin indicate the effect of endurance training on intact vs. traumatized tissue (**C**). White arrows show colocalization of APC with DAPI. Results are presented as mean ± SD (RT-PCR: *n* = 4, WB: *n* = 5). Data were statistically evaluated using one-way ANOVA and post hoc Tukey’s HSD test; * *p* < 0.05; ** *p* < 0.01; *** *p* < 0.001. Scale bar: 100 µm.

**Figure 6 ijms-24-08616-f006:**
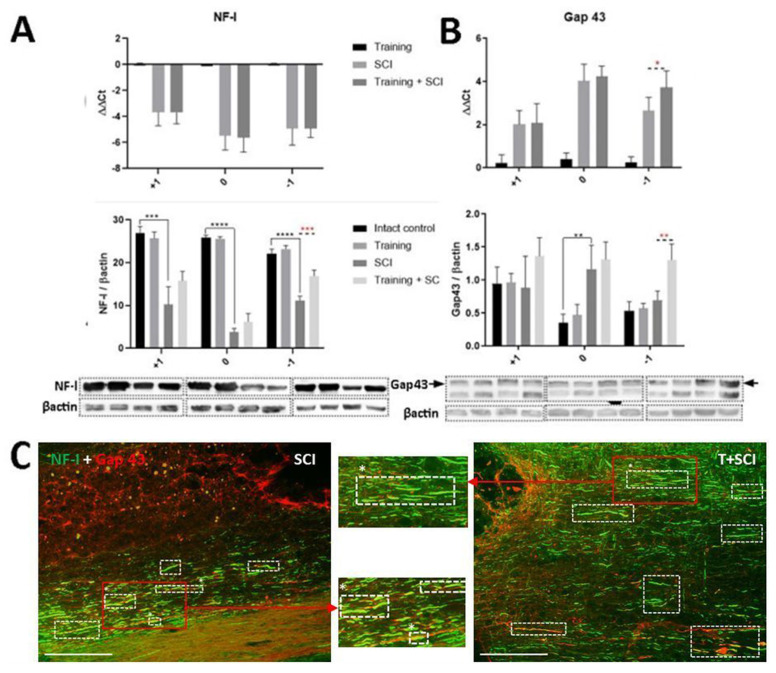
Gene expression, protein level analysis, and fluorescent labeling of light chains of neurofilaments (NF-l) and newly outgrowing fibers (Gap-43) in experimental groups. Levels of ΔΔCt for NF-l, Gap-43, and the ratio of NF-l/βactin and Gap-43/βactin show the effect of endurance training on intact vs. traumatized spinal cord tissue (**A**,**B**) (RT-PCR: *n* = 4, WB: *n* = 5). Microphotographs show the staining of NF-l (green) and Gap-43 (red) in SCI and T + SCI groups in caudal segment (−1) (**C**) (*n* = 3). Red arrows indicate the magnification of representative dashed white squares, showing colocalization of NF-1 with Gap-43. Results are presented as mean ± SD. Data were statistically evaluated using one-way ANOVA and post hoc Tukey’s HSD test; * *p* < 0.05; ** *p* < 0.01; *** *p* < 0.001; **** *p* < 0.0001. Scale bar: 100 µm.

**Figure 7 ijms-24-08616-f007:**
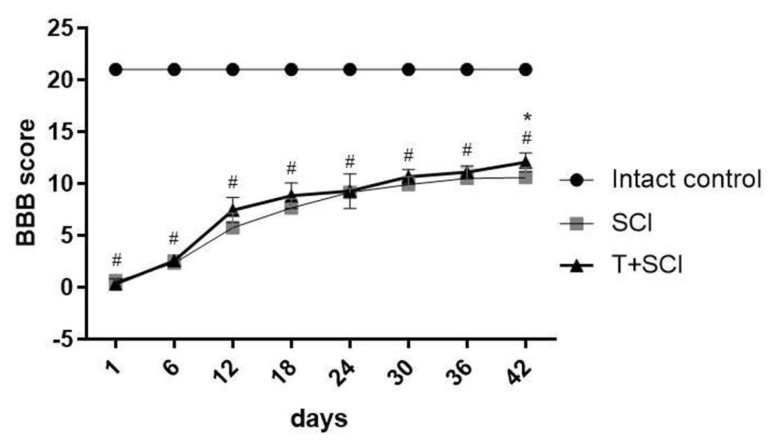
Neurological Basso–Beatie–Bresnahan (BBB) score showing locomotor activity of experimental animals in intact control (IC), SCI and T + SCI groups. Scoring ranges from 0 points (complete paraplegia) to 21 points (normal hindlimb movements). Results are presented as mean ± SD (*n* = 9). Data were statistically evaluated using repeated measures of variance (repeated one-way ANOVA multiple comparisons); experimental groups vs IC (^#^ *p* < 0.05); SCI vs T + SCI (* *p* < 0.05).

**Figure 8 ijms-24-08616-f008:**
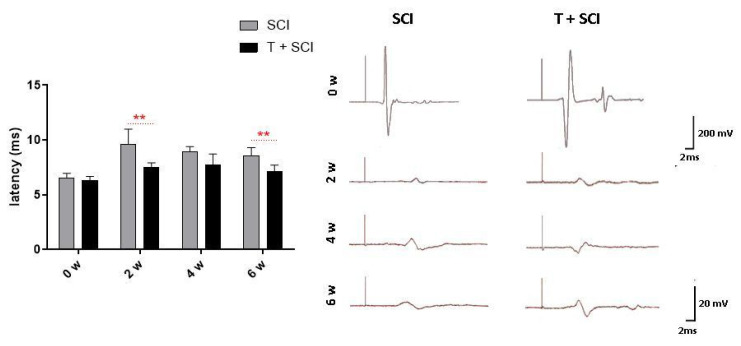
Values of peak latency (ms) recorded from musculus gastrocnemius prior to spinal cord injury (0 weeks) and 2, 4, and 6 weeks after trauma in the SCI and T + SCI groups. Results are presented as mean ± SD (*n* = 4). Data were statistically evaluated using Student’s parametric *t*-test; ** *p* < 0.01.

**Figure 9 ijms-24-08616-f009:**
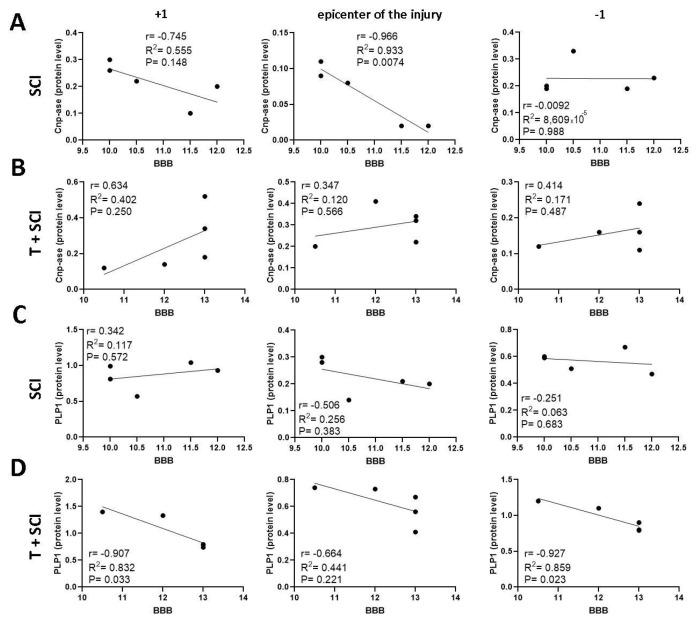
Linear regression analysis of correlation between protein levels for CNP-ase (**A**,**B**), PLP1 **(C,D)**, and BBB neurological score (**A**–**D**) in the epicenter of injury and adjacent segments in SCI and T + SCI groups. Scatterplots of individual values with regression line correlation coefficient (r) and coefficients of determination (R^2^) calculated using regression analysis and *p*-Value (*n* = 5).

**Figure 10 ijms-24-08616-f010:**
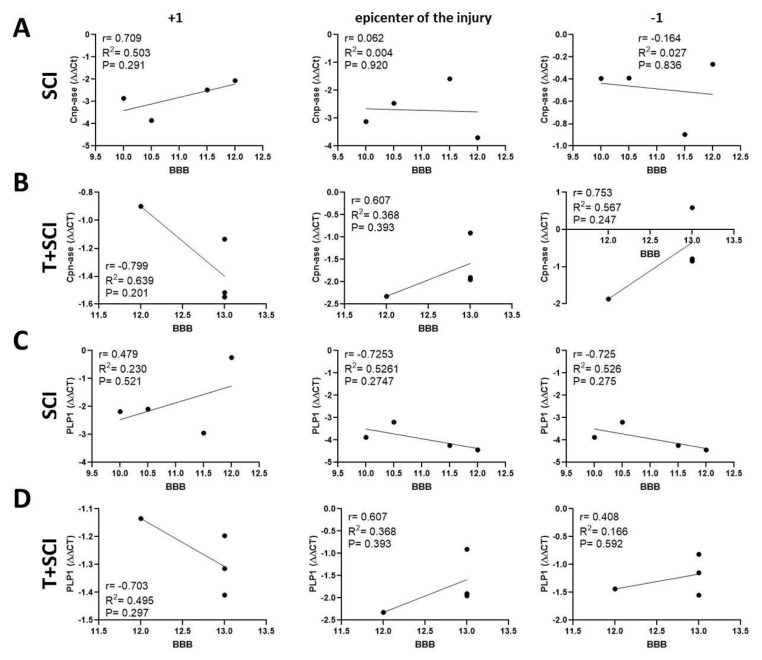
Linear regression analysis of correlation between gene expression for CNP-ase (**A**,**B**), PLP1 (**C**,**D**), and BBB neurological score (**A**–**D**) in the epicenter of injury and adjacent segments in SCI and T + SCI groups. Scatterplots of individual values with regression line correlation coefficient (r) and coefficients of determination (R^2^) calculated using regression analysis and *p*-Value (*n* = 4).

**Figure 11 ijms-24-08616-f011:**
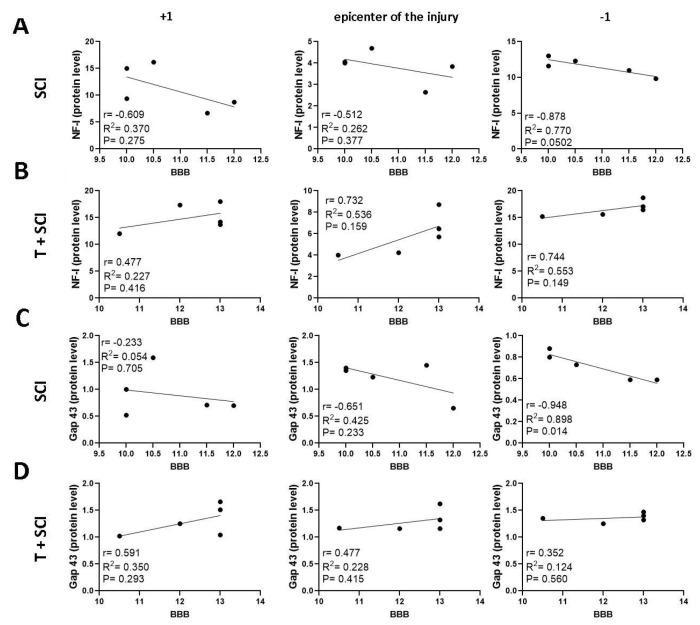
Linear regression analysis of correlation between protein levels for NF-l (**A**,**B**), Gap-43 (**C**,**D**), and BBB neurological score (**A**–**D**) at the lesion site, cranial, and caudal segments in Th9 compression groups (SCI and T + SCI). Scatterplots of individual values with regression line correlation coefficient (r) and coefficients of determination (R^2^) calculated using regression analysis and *p*-Value (*n* = 5).

**Figure 12 ijms-24-08616-f012:**
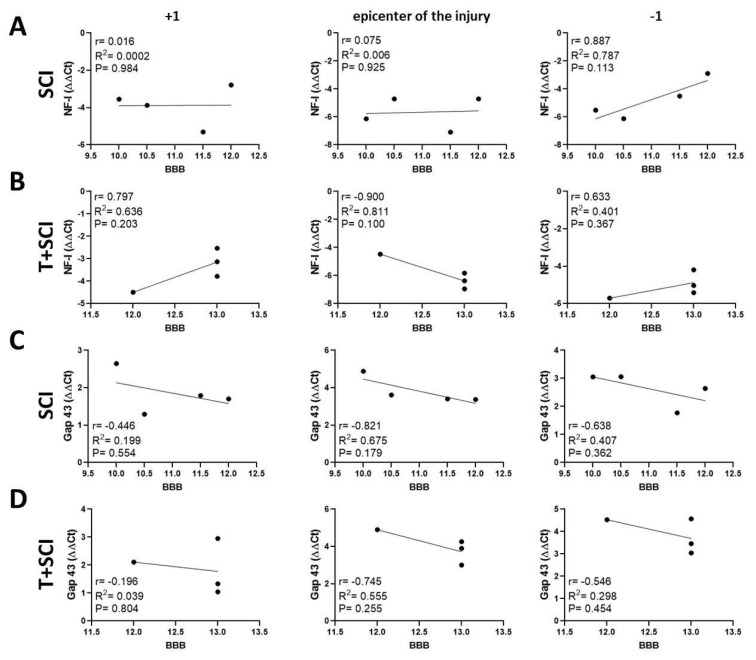
Linear regression analysis of correlation between gene expression for NF-l (**A**,**B**), Gap-43 (**C**,**D**), and BBB neurological score (**A**–**D**) at the lesion site, cranial, and caudal segments in Th9 compression groups (SCI and T + SCI). Scatterplots of individual values with regression line correlation coefficient (r) and coefficients of determination (R^2^) calculated using regression analysis and *p*-Value (*n* = 4).

**Figure 13 ijms-24-08616-f013:**
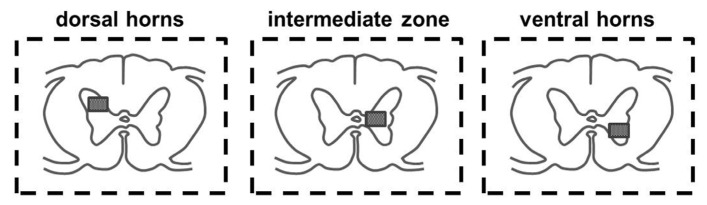
Schematic illustration of the spinal cord showing localization of three areas of interest (dorsal horns, intermediate zone, and ventral horns) for quantitative analysis of GAD 65- and Glyt2-positive cells. Measuring frame size in each area was 1.1 × 1 mm.

**Table 1 ijms-24-08616-t001:** Number of GABA-ergic (GAD 65) and glycinergic (Glyt2) neurons in dorsal horns, intermediate zone, and ventral horns of Th10 spinal segment. The quantity of inhibitory neurons is shown in intact control (IC), trained group (T), after Th9 compression (SCI), and after training followed by Th9 compression (T + SCI). Results are presented as mean ± SD. Data were statistically evaluated using one-way ANOVA and post hoc Tukey’s HSD test; * *p* < 0.05; ** *p* < 0.01; *** *p* < 0.001; ns = nonsignificant.

			Significance	
		IC vs. T	IC vs. SCI	SCI vs. T + SCI
GABA (Th10 segment)	IC	T	SCI	T + SCI
dorsal horns	0.70 ± 0.83	0.36 ± 0.67	0 ± 0	0.58 ± 0.99
		ns	**	*
intermediate zone	4.50 ± 1.65	4.10 ± 1.66	1.84 ± 1.28	2.10 ± 1.66
		ns	***	ns
ventral horns	6.25 ± 2.09	6.00 ± 1.83	4.35 ± 1.08	4.41 ± 1.51
		ns	**	ns
Glyt2 (Th10 segment)	IC	T	SCI	T + SCI
dorsal horns	0.10 ± 0.31	1.81 ± 1.60	0.17 ± 0.38	0.45 ± 0.68
		**	ns	ns
intermediate zone	4.00 ± 1.63	4.63 ± 1.43	2.91 ± 1.22	3.00 ± 1.10
		ns	ns	ns
ventral horns	6.50 ± 1.72	7.27 ± 1.95	4.83 ± 1.53	5.09 ± 1.14
		ns	*	ns

**Table 2 ijms-24-08616-t002:** Number of GABA-ergic (GAD 65) and glycinergic (Glyt2) neurons in dorsal horns, intermediate zone, and ventral horns of L2 spinal segment. The quantity of inhibitory neurons is shown in intact control (IC), trained group (T), after Th9 compression (SCI), and after training followed by Th9 compression (T + SCI). Results are presented as mean ± SD. Data were statistically evaluated using one-way ANOVA and post hoc Tukey’s HSD test; * *p* < 0.05; ** *p* < 0.01; *** *p* < 0.001; ns = nonsignificant.

			Significance	
		IC vs. T	IC vs. SCI	SCI vs. T + SCI
GABA (L2 segment)	IC	T	SCI	T + SCI
dorsal horns	1.38 ± 0.91	0.44 ± 0.72	0.63 ± 1.21	0.60 ± 0.69
		*	ns	ns
intermediate zone	5.62 ± 1.92	4.37 ± 0.91	4.00 ± 2.00	5.50 ± 1.43
		ns	ns	ns
ventral horns	11.13 ± 1.35	8.77 ± 1.39	9.09 ± 1.57	9.70 ± 1.83
		**	**	ns
Glyt2 (L2 segment)	IC	T	SCI	T + SCI
dorsal horns	0.87 ± 1.25	0.55 ± 0.72	0.33 ± 0.71	1.13 ± 1.64
		ns	ns	ns
intermediate zone	4.25 ± 1.66	3.77 ± 7.09	1.56 ± 1.51	5.13 ± 1.46
		ns	**	***
ventral horns	10.67 ± 3.12	8.40 ± 2.45	7.44 ± 1.23	10.00 ± 1.93
		ns	*	**

**Table 3 ijms-24-08616-t003:** List of genes and primer nucleotide sequences used in RT-PCR study.

Targets	Forward Primer	Reverse Primer
CNP-ase	CCTCAGAGCCACCACACATC	CCTTCCTTGGGGCTACAGTG
PLP1	TTGGCGACTACAAGACCACC	AATGACACACCCGCTCCAAA
NF-l	AAGGCTAAGACCCTGGAGATCGAAG	GGGATAGTTGGGAATGGGGCTCAA
Gap-43	AGGAAAGGAGAGAAGGCAGG	GCAGGAGAGACAGGGTTCAG

## Data Availability

The datasets generated and analyzed during the current study are available from the corresponding author on reasonable request.

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
