# Peer review of "Impact of Endurance Training on Regeneration of Axons, Glial Cells, and Inhibitory Neurons after Spinal Cord Injury: A Link between Functional Outcome and Regeneration Potential within the Lesion Site and in Adjacent Spinal Cord Tissue"

_ijms, 2023, doi:10.3390/ijms24108616_

Round 1
Reviewer 1 Report
Dear authors. The paper submitted for review is very well prepared. The research performed is well done, the results presented are correctly described, statistical analysis has been performed. Citations are used correctly in the paper, the paper is well written.
Unfortunately, after reading the paper I have a fundamental question about the meaning of the research. Therefore, please answer the following questions:
1. will the knowledge gained from the research help in the treatment of spinal cord injuries.
2. whether the information obtained will allow the development of, for example, new effective cellular therapies.
3. whether the findings will reduce the severity of spinal cord injuries.
4. whether the authors have information on whether, for example, the results of the research obtained correlate with the clinical picture in sports people.
Please also consider adding these questions to the paper - introduction and/or discussion.
Reviewer 2 Report
This is a very interesting study concerning the endurance training prior to spinal cord injury to exert the protective effect. This effect was mainly due to gene expression and protein level of immature CNP-ase in oligoden-drocytes upon the injured region and neurotrophic regulation in the inhibitory GABA/glycinergic neurons adjacent to the injure area. This study was well written and but not helpful for the readers.
In the clinical cases, the spinal cord injury was incidental. The authors recommend the pre-injury endurance training to reduce the spinal cord injury, which seems not to be logical and cannot be helpful in real word. The authors should provide the sound evidence to prove this hypothesis.
Round 2
Reviewer 1 Report
Dear Authors. Thank you for the amendments brought, unfortunately, I have not received answers to the questions. The work accepts.